# Behavioural response to the Covid-19 pandemic in South Africa

Umakrishnan Kollamparambil●*◉, Adeola Oyenubi●◉

School of Economics and Finance, University of the Witwatersrand, Johannesburg, South Africa

◉ These authors contributed equally to this work.
* uma.kollamparambil@wits.ac.za

## Abstract

### Background

Given the economic and social divide that exists in South Africa, it is critical to manage the health response of its residents to the Covid-19 pandemic within the different socio-economic contexts that define the lived realities of individuals.

### Objective

The objective of this study is to analyse the Covid-19 preventive behaviour and the socio-economic drivers behind the health-response behaviour.

### Data

The study employs data from waves 1 and 2 of South Africa's nationally representative National Income Dynamics Study (NIDS)—Coronavirus Rapid Mobile Survey (CRAM). The nationally representative panel data has a sample of 7073 individuals in Wave 1 and 5676 individuals in Wave 2.

### Methods

The study uses bivariate statistics, concentration indices and multivariate estimation techniques, ranging from a probit, control-function approach, special-regressor method and seemingly unrelated regression to account for endogeneity while identifying the drivers of the response behaviour.

### Findings

The findings indicate enhanced behavioural responsiveness to Covid-19. Preventive behaviour is evolving over time; the use of face mask has overtaken handwashing as the most utilised preventive measure. Other measures, like social distancing, avoiding close contact, avoiding big groups and staying at home, have declined between the two periods of the study. There is increased risk perception with significant concentration among the higher income groups, the educated and older respondents. Our findings validate the health-belief model, with perceived risk, self-efficacy, perceived awareness and barriers to preventive strategy adoption identified as significant drivers of health-response behaviour. Measures

**Data Availability Statement:** Data are available: http://www.nids.uct.ac.za/nids-cram/data-access.

**Funding:** This study was funded by The CRAM Project. The funders had no role in study design,

data collection and analysis, decision to publish, or preparation of the manuscript.

**Competing interests:** The authors have declared that no competing interests exist.

such as social distancing, avoiding close contact, and the use of sanitisers are practised more by the rich and educated, but not by the low-income respondents.

## Conclusion

The respondents from lower socio-economic backgrounds are associated with optimism bias and face barriers to the adoption of preventive strategies. This requires targeted policy attention in order to make response behaviour effective.

## 1. Introduction

The impact of the corona virus pandemic on the South African economy and the health of its residents is evolving in real time. South Africa went into a hard lockdown (Level 5) quite early in the pandemic (March 2020). During the hard lockdown, the residents were mostly confined to their homes, leaving very little need for proactive decision-making by individuals. The complete lockdown, however, was untenable, even with the fiscal relief measures put in place by the government [1]. The economic implications of the nationwide shutdown made it unsustainable, with increasing levels of hunger, poverty and unemployment among the vulnerable sections of society [2].

Since then, the government has reduced the levels of lockdown restrictions in phases to permit the economy to function once again. The government declared the move to lockdown Level 4 from 1 May, Level 3 from 1 June and, Level 2 from 18 August. Behavioural restrictions have been lifted in a calibrated manner commensurate with the lockdown level, albeit with precautionary messaging from the government. While regulations have been passed making the wearing of face masks essential, the limited capacity for monitoring implies that it is largely left to the individuals to comply with the regulation and other precautionary measures to prevent Covid-19 infection. This means that the responsibility of managing the pandemic through restrained behaviour has essentially shifted to the residents of the country. With no immediate prospects of eradicating Covid-19, non-pharmaceutical interventions remain the most effective defence against the pandemic [3].

Therefore, the control of the pandemic depends on the behavioural response of individuals. Findings from the NIDS-CRAM Wave 1, however, suggest that high-impact behaviour changes are not happening fast enough in South Africa, even though a large percentage of the population reported some form of change in behaviour [4]. In a country as economically and socially divided as South Africa [5, 6], it would be unrealistic to expect a uniform response from its residents. The purpose of this policy paper is to explore the role of socio-economic contexts in determining the health-behaviour response of individuals. The findings of this study will enable effective policymaking by taking into account socio-economic inequality in behavioural responses. The purpose of this study is to provide this perspective to policy makers that will enable a more nuanced policy formulation.

The study seeks answers to the following questions:

1. How has preventive behaviour evolved over the ongoing Covid-19 pandemic?

2. What are the key drivers behind the health-response behaviour?

The study uses bivariate statistics to identify significant differences across binary variables like sex, race, age and geographical location. Concentration indices are used to estimate the

income, education and age-related inequalities in behavioural response. Lastly, multivariate regression analysis is used to identify the drivers of the response behaviour.

## 2. Literature review

The study adopts the Health Belief Model (HBM) as its analytical framework. The Health Belief Model [7, 8] highlights the role of people's awareness, perceived risk, self- efficacy (the confidence and belief that pro-health action can yield desirable outcomes), and feasibility of precautionary/preventive action in explaining individual health-response behaviour. In a country like South Africa that is torn apart by dual realities (South Africa has is one of the most unequal countries in the world with an income related Gini coefficient of over 0.63 (Kollamparambil 2020a)), it is important not to assume a common behavioural response from all sections of society. The HBM model is therefore suited to understanding the differential responses from individuals in a country that is known as one of the most unequal in the world.

The available literature on the individual protective-response behaviour to a pandemic (prior to Covid-19) is within the context of the HINI virus and the SARS virus. [9] provide an effective review of some of the most pertinent studies that look into the demographic and attitudinal determinants of protective behaviours during a pandemic. The review identified 26 papers that met the study inclusion criteria. The studies were of variable quality and most lacked an explicit theoretical framework. With the exception of [10], others were cross-sectional in design, with little predictive power over time. The research shows that there are demographic differences in behaviour: being older, female and more educated, or non-White, is associated with a higher chance of adopting the behaviours. There is evidence that greater levels of perceived susceptibility to and perceived severity of the diseases and greater belief in the effectiveness of recommended behaviours to protect against the disease are important predictors of behaviour. There is also evidence that greater levels of state anxiety and greater trust in authorities are associated with preventive behaviour. The findings are, however, questionable considering that the endogeneity of variables measuring risk perceptions have seldom been acknowledged, let alone accounted for, in empirical analyses. This is despite the simultaneity of the relationship between risk perception and health behaviours being acknowledged in other contexts ([11] for HIV/AIDS and sexual behaviour, [12] for drug safety warnings and preventive use, among others). In this article, we argue that risk perceptions have to be treated as endogenous to preventive behaviour in order to produce accurate and reliable measures of an individual's valuation of the pandemic risk.

Literature on behavioural response to Covid-19 is fast emerging [13–18]. The limitations highlighted in the context of the pandemic literature prior to Covid-19 holds true for the more recent studies as well. The studies are mostly cross-sectional descriptive statistics analyses within developed country contexts. In the absence of a multivariate analysis, the studies do not account for correlations between the various factors and therefore confuse mediating variables with the real drivers of behaviour.

[19] is one of the earliest studies undertaken in the context of Poland, reporting the role of dark personality traits as drivers of behaviour during the pandemic. The study, however, did not account for socio-economic and demographic characteristics. Another key limitation is the sample size of the study, which is below 900 for both waves. Further, the two waves were conducted within a gap of two weeks, reducing its predictive power over time.

The relevance of the Health Belief Model in explaining the behavioural change within the context of the Covid-19 pandemic is also established by [20]. The small sample and limited geographical coverage of the cross-sectional survey held within the Kerala State of India provides little external validity. Further, although the study undertakes a logistic regression, it

does not account for endogeneity. Similar limitations exist with the other attempts in other country contexts ([21] for South Korea and [22] for the US).

The current study is one of the first on the Covid-19 pandemic that is based on nationally representative data within the context of an African country and contributes to the Health Belief literature by building in an estimation strategy that accounts for endogeneity in the model.

## 3. Data

### 3.1 Summary statistics

The analysis utilises the first and second waves of the National Income Dynamics Survey (NIDS)—Coronavirus Rapid Mobile Survey (CRAM) [23, 24]. The NIDS-CRAM survey is a special follow up with a subsample of adults from households in Wave 5 of the National Income Dynamics Study (NIDS) run by SALDRU [25]. The NIDS-CRAM has a smaller sample by comparison, covering complete questionnaire information for 7073 individuals in Wave 1 and 5676 individuals in Wave 2. The survey is designed to be nationally representative and remains the best available source of quantitative information on a national scale to assess the socio-economic impact of the corona virus pandemic in South Africa.

The NIDS-CRAM sample is drawn using a stratified sampling design with "batch sampling", by which the sampled individuals were sent to the fieldwork team in batches of 2500 individuals [26]. The response rate in NIDS-CRAM was approximately 40%. The sampling process incorporated a non-response adjustment by oversampling strata where strata response rates in the initial batches were low. A further 8% of the selected respondents were classified as a refusal; they were contacted but refused to be interviewed. The non-response adjustment is undertaken following [27], whereby the design weight is multiplied by the inverse of the conditional probability of being interviewed. Further, trimming is used to adjust the weights with weights below the 1st percentile of all weight values set to the 1st percentile and those weights above the 99th percentile set to the 99th percentile [27]. Lastly, the issue of panel attrition between waves 1 and 2 is addressed by [28]. Using probit regression models to predict the determinants of attrition, the study found attrition to be random across all model specifications.

The first wave of the NIDS-CRAM survey was conducted over the months of May and June 2020, and the second wave was administered in July and August 2020. Therefore, about 25% of the first wave was under lockdown Level 4 conditions, while 75% was under Level 3 conditions. The second wave has been conducted in its entirety over lockdown Level 3 conditions. Where possible, the analysis is structured to explore the evolving behaviour using both waves for comparison. However, this is limited by the availability of information in both the survey waves. For example, certain variables like the sources of information about Covid were collected only in the first wave and therefore the analysis relies exclusively on the first wave for it.

In order to minimise the pressure to respond in a socially desirable way, all survey participants were verbally informed that their identity would be kept confidential and all the collected information would be anonymised. Further, the respondents were made aware that participation in the study was voluntary, and that they could stop the interview at any time. Despite all of these measures, it is hard to assert that there was no strategic bias. The study therefore acknowledges the limitation that the analysis is based on self-reported data and therefore susceptible to hypothetical and strategic bias [29]. A further limitation to be highlighted is the high proportion of missing information on household income. The analysis relating to household income therefore is restricted to 3599 individuals in Wave 1 and 3569 individuals in Wave 2. Despite these challenges, the NIDS-CRAM survey remains the best available source of

**Table 1. Descriptive statistics of the key socio-economic variables.**

| Variables | Wave 1 | | | Wave 2 | | | |
|---|---|---|---|---|---|---|---|
| | Mean/ % | Lower conf interval | Upper conf interval | Mean/ % | Lower conf interval | Upper conf interval | Population (2011 census) |
| Male (%) | 46. 9 | | | 46.8 | | | 48.7 |
| Female (%) | 53.1 | | | 53.2 | | | 51.4 |
| Black (%) | 78.7 | | | 78.7 | | | 79.2 |
| Non-Black (%) | 21.3 | | | 21.3 | | | 20.8 |
| Urban (%) | 81.9 | | | 75.8 | | | 66 |
| Rural (%) | 18.1 | | | 24.2 | | | 44 |
| Education less than 8 years | 18.7 | | | 18.9 | | | 19.1 |
| Age years | 40.1 | 39.5 | 40.8 | 40.3 | 39.6 | 40.9 | 25 |
| HH income (Rands) | 87768 | 81570.67 | 93958.56 | 83368 | 78112.9 | 88623.5 | 103 204* |
| Income Quintiles (Rands) | | | | | | | |
| 1 | 97.479 | 80.594 | 114.364 | 153.756 | 132.543 | 174.969 | |
| 2 | 404.922 | 365.610 | 444.235 | 409.526 | 378.265 | 440.788 | |
| 3 | 720.599 | 655.520 | 785.679 | 758.305 | 647.095 | 869.515 | |
| 4 | 1431.516 | 1284.740 | 1578.292 | 1461.341 | 1300.360 | 1622.321 | |
| 5 | 8626.576 | 7371.994 | 9881.158 | 8255.206 | 6831.270 | 9679.141 | |

Source: NIDS-CRAM, Wave 1 (2020) & NIDS-CRAM, Wave 2 (2020)

* https://www.statssa.gov.za/publications/P03014/P030142011.pdf

Notes: Data are weighted.

data to analyse the nation's response to the pandemic. The descriptive statistics of the key socio-economic variables in the sample (Table 1) indicate resonance with national statistics.

Table 1 shows that the two samples are fairly representative of the national population (based on the 2011 census) in terms of the important demographic variables such as race, sex, and education. As the sample is restricted to adults, the average sample age is higher than that of the population. The reduction in average income observed in the sample is in line with the economic devastation that occurred during the pandemic.

## 3.2 Health-response behaviour

It is reassuring to see that there has been significant improvement in the percentage of individuals reporting adopting some form of behavioural change in response to the threat of corona virus. While 92% reported changing their behaviour in Wave 1, this rose to 99.7% in Wave 2 (Fig 1).

There are significant changes in the preventive measures used between the two waves (Fig 2). While in Wave 1, handwashing was the predominant measure, this has changed to the use of face masks in Wave 2. It is clear the individuals are responding to public messaging; in the initial phases handwashing was emphasised over face mask use. Subsequently, the expert views available to the public changed in favour of face masks and this is reflected in the surge of face mask use from under 50% to over 70%. While this is heartening, it needs to be noted that a significant proportion is still not utilising face masks despite the regulation making it mandatory.

The other major shift observed in preventive behaviour is the reduction in the practice of physical distancing (Fig 2). With the opening up of the economy, it is noticeable that staying at home has reduced substantially from just under 50% to well below 40%. More concerning is that those reporting social distancing, avoiding close contact and avoiding big groups have reduced significantly. While this seems to be compensated for through the increased use of

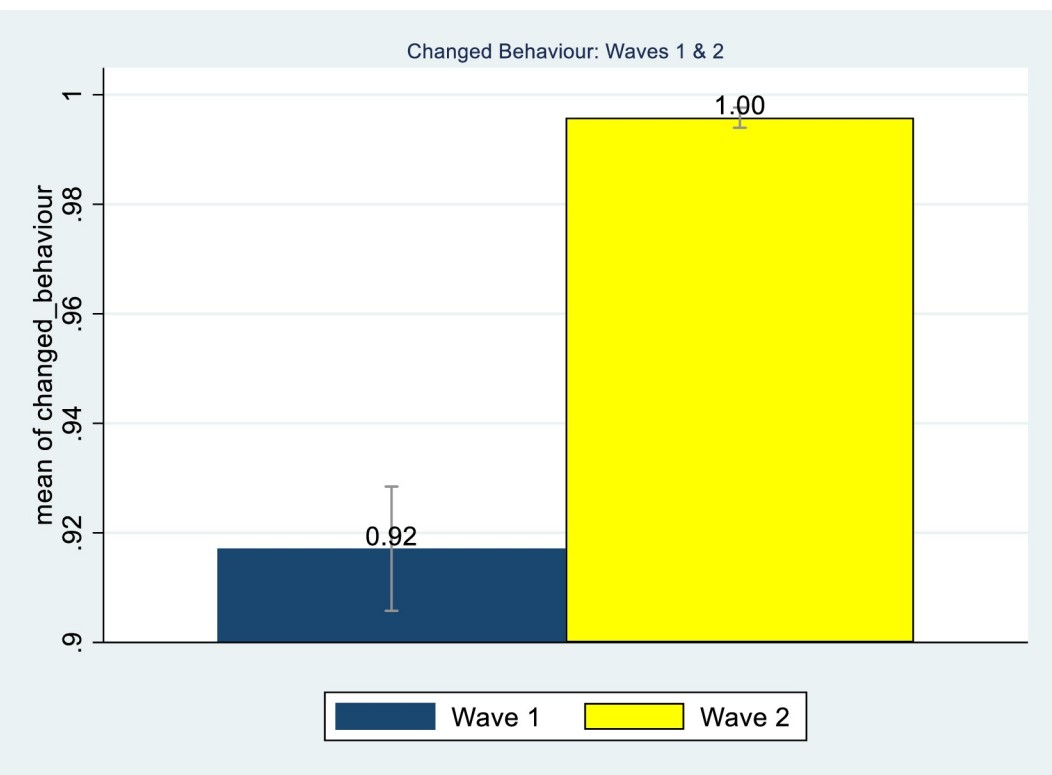

**Fig 1. Behavioural change across Wave 1 and Wave 2.** Source: NIDS-CRAM, Wave 1 (2020) and NIDS-CRAM, Wave 2 (2020). Data are weighted. The bars show the reported change in behaviour. 95% confidence intervals are shown.

face masks and hand sanitisers, the reduction in physical distancing measures remains a concern.

The socio-economic inequality in the use of preventive measures is revealing. Awareness of the necessary measure is a necessary precondition, but the barriers to adopting it within an

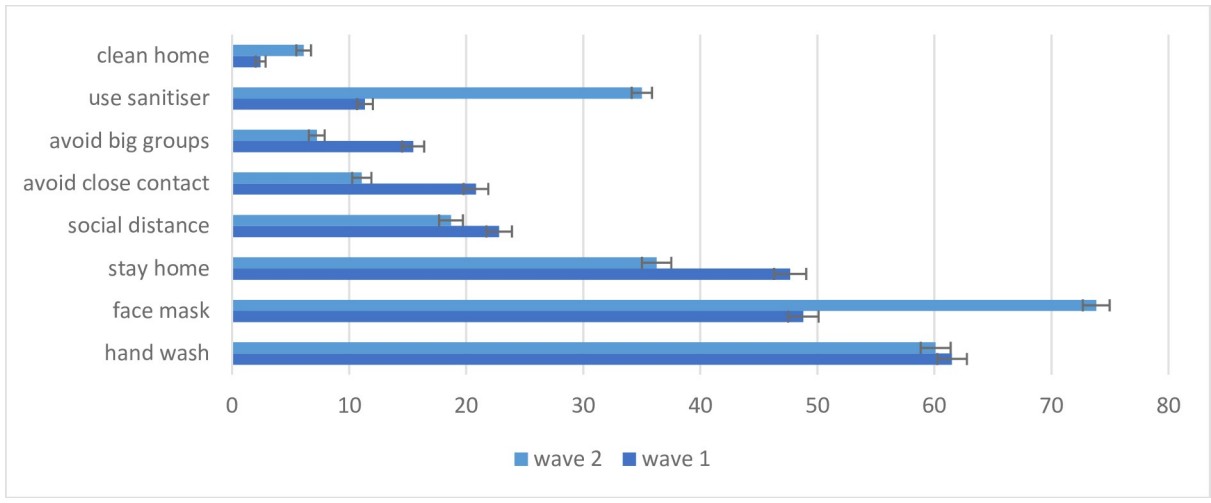

**Fig 2. Types of preventive measures.** Source: NIDS-CRAM, Wave 1 (2020) and NIDS-CRAM, Wave 2 (2020). Data are weighted. The bars show the use of different preventive measures in percentage. 95% confidence intervals are shown.

**Table 2. Erreygers'-corrected concentration index (income-related) of preventive measures.**

| Income-Related | Concentration Index Wave 1 | Lower Bound | Upper Bound | Concentration Index Wave 2 | Lower Bound | Upper Bound |
|---|---|---|---|---|---|---|
| Changed behaviour | 0.005 | -0.025 | 0.035 | -0.003 | -0.006 | 0.000 |
| Handwash | -0.033 | -0.095 | 0.030 | -0.039 | -0.104 | 0.026 |
| Avoid close contact | **0.095**\*\*\* | 0.038 | 0.151 | **0.040**\*\* | 0.002 | 0.079 |
| Avoid big groups | 0.056 | -0.002 | 0.115 | -0.002 | -0.037 | 0.034 |
| Face mask | **0.088**\*\*\* | 0.024 | 0.152 | 0.019 | -0.037 | 0.074 |
| Stay home | **-0.088**\*\*\* | -0.155 | -0.021 | -0.122 | -0.183 | -0.061 |
| Use sanitiser | 0.042 | -0.006 | 0.090 | **0.105**\*\*\* | 0.041 | 0.169 |
| Clean home | -0.002 | -0.025 | 0.020 | 0.040\*\* | 0.011 | 0.069 |
| Social distance | **0.106**\*\*\* | 0.049 | 0.163 | **0.067**\*\* | 0.011 | 0.123 |
| Flu vaccine | 0.004 | -0.016 | 0.023 | 0.008 | -0.001 | 0.017 |

Source: NIDS-CRAM, Wave 1 (2020) and NIDS-CRAM, Wave 2 (2020)

Notes: Data are weighted. CI is the confidence interval.

\*\*\* $p<0.01$

\*\* $p<0.05$

\* $p<0.1$

individual's living and livelihood conditions might make it infeasible. Therefore, socio-economic context is expected to play a key role in the nature of preventive measures adopted by individuals. While in Wave 1, the use of face masks was concentrated among the economically affluent, the concentration index is not statistically significant in Wave 2 (Table 2). This is an encouraging sign, that face mask use has spread across income groups. However, physical distancing practices like social distancing and avoiding close contact remains pro-rich. Or in other words, these practices are significantly more concentrated among the rich than the poor. This highlights the question of the feasibility of these preventive measures for respondents who live in crowded households and neighbourhoods and have no alternative to public transport. The lifting of capacity restrictions in public taxis have particular bearing for individuals from the lower economic strata of society, exposing them to higher risks compared to those with private transport. Staying home as a preventive measure is seen to be concentrated among the poor but is not significant in Wave 2.

Education-related inequality is visible along the lines of the income-related inequality in the use of preventive measures (Table 3). The results indicate that social distancing, avoiding close contact and the use of sanitisers are practised more among the educated. The use of face masks was also pro-educated in Wave 1 but has become insignificant in Wave 2, indicating its popularity cutting across education lines.

Age related concentration indices indicate that the concentration in behavioural change amongst the young has declined, but still remains significant (Table 4). While the use of a flu vaccine is revealed as a strategy concentrated amongst the older individuals, social distancing is evident as a measure concentrated amongst the younger individuals.

### 3.3 Risk perception

Individual risk perception is an important pillar in the Health Belief model. The process of individual risk perception determination is seldom entirely rational. Both cognitive and emotional assessments contribute to the formulation of risk perception. While cognitive skills through the logical weighing of evidence and reasoning contribute to risk perception formulation, equally, emotional appraisals, through the use of intuition and imagination, play an

**Table 3. Erreygers'-corrected concentration index (education-related) of preventive measures.**

| Education-related | Concentration Index Wave 1 | Lower Bound CI | Upper Bound CI | Concentration Index Wave 2 | Lower Bound CI | Upper Bound CI |
|---|---|---|---|---|---|---|
| Changed behaviour | 0.005 | -0.025 | 0.035 | -0.003 | -0.006 | 0.000 |
| Handwash | -0.033 | -0.095 | 0.030 | -0.039 | -0.104 | 0.026 |
| Avoid close contact | **0.095**\*\*\* | 0.038 | 0.151 | **0.040**\*\* | 0.002 | 0.079 |
| Avoid big groups | 0.056 | -0.002 | 0.115 | -0.002 | -0.037 | 0.034 |
| Face mask | **0.088**\*\*\* | 0.024 | 0.152 | 0.019 | -0.037 | 0.074 |
| Stay home | **-0.088**\*\*\* | -0.155 | -0.021 | -0.122 | -0.183 | -0.061 |
| Use sanitiser | 0.042 | -0.006 | 0.090 | **0.105**\*\*\* | 0.041 | 0.169 |
| Clean home | -0.002 | -0.025 | 0.020 | **0.040**\*\* | 0.011 | 0.069 |
| Social distance | **0.106**\*\*\* | 0.049 | 0.163 | **0.067**\*\* | 0.011 | 0.123 |
| Flu vaccine | 0.004 | -0.016 | 0.023 | 0.008 | -0.001 | 0.017 |

Source: NIDS-CRAM, Wave 1 (2020) and NIDS-CRAM, Wave 2 (2020)

Notes: Data are weighted. CI is the confidence interval.

\*\*\* $p<0.01$

\*\* $p<0.05$

\* $p<0.1$

important role [30]. The literature has highlighted the role of optimism bias (the tendency to believe that one's own risk is less than that of others) in reducing the health-protective behaviour or increasing risk-taking [31]. It is therefore important to identify the high-risk taking category for targeted policymaking.

The risk perception information in the study was obtained through the 'yes' or 'no' response to the question 'Do you think you are likely to get the corona virus?'. As indicated earlier, just over 75% of responses in Wave 1 were captured under lockdown Level 3 and the rest under lockdown Level 4 and 100% of Wave 2 responses were sought in Level 3 conditions. Despite this, the findings show that there was a significant increase in risk perception in Wave 2 relative to Wave 1. While 33% of individuals reported a risk of infection, this increased to 50% in Wave 2 (Fig 3).

**Table 4. Erreygers'-corrected concentration index (age-related) of preventive measures.**

| Age-related | Concentration Index Wave 1 | Lower Bound CI | Upper Bound CI | Concentration Index Wave 2 | Lower Bound CI | Upper Bound CI |
|---|---|---|---|---|---|---|
| Changed behaviour | **-0.041**\*\*\* | -0.072 | -0.010 | **-0.006**\*\* | -0.011 | -0.001 |
| Handwash | **-0.051**\* | -0.102 | 0.001 | -0.023 | -0.071 | 0.025 |
| Avoid close contact | -0.014 | -0.050 | 0.022 | 0.002 | -0.031 | 0.036 |
| Avoid big groups | -0.002 | -0.038 | 0.034 | -0.005 | -0.027 | 0.018 |
| Face mask | -0.002 | -0.048 | 0.043 | -0.020 | -0.064 | 0.025 |
| Stay home | 0.000 | -0.010 | 0.010 | 0.001 | -0.004 | 0.006 |
| Use sanitiser | 0.006 | -0.007 | 0.018 | -0.010 | -0.025 | 0.005 |
| Clean home | -0.024 | -0.080 | 0.032 | 0.027 | -0.024 | 0.078 |
| Social distance | -0.004 | -0.033 | 0.024 | **-0.141**\*\*\* | -0.189 | -0.093 |
| Flu vaccine | 0.002 | -0.004 | 0.008 | **0.002**\*\* | 0.000 | 0.005 |

Source: NIDS-CRAM, Wave 1 (2020) and NIDS-CRAM, Wave 2 (2020)

Notes: Data are weighted. CI is confidence interval.

\*\*\* $p<0.01$

\*\* $p<0.05$

\* $p<0.1$

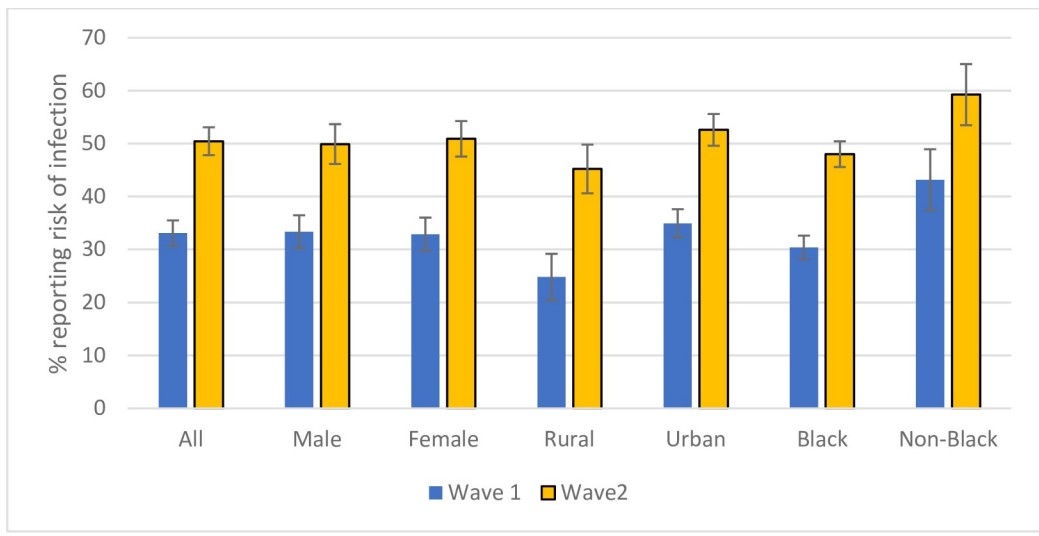

**Fig 3. Covid-19 risk perception over Wave 1 and Wave 2.** Source: NIDS-CRAM, Wave 1 (2020) and NIDS-CRAM, Wave 2 (2020) Data are weighted. The bars show the risk perception in percentage. 95% confidence intervals are shown.

Emerging research on the vulnerability to Covid-19 infection has highlighted the role of covariates associated with poverty. These include the non-availability of a private mode of transport, lack of access to information, lack of hygiene facilities at home like water and sanitation, over-crowded households, and multi-generational households [32]. Therefore, considering the race-based poverty-rate differences in South Africa [6], it is surprising to note that the South African black population group perceive a significantly lower risk (48% in Wave 2) compared to non-blacks (60% in Wave 2). Even though there is a significant increase in the risk perception of black Africans in the second wave, the non-black risk perception has also increased and therefore the race gap remains significant in both waves. The gap between the black and non-black categories could be interpreted as the result of a dual bias. The high risk-perception of non-blacks (with lower poverty rates) is due to an over assessment of risk and optimism bias on the part of the black African population (with higher poverty rate). Together, both biases compound to create a significant gap in risk perception between black and non-black population groups.

A comparison of the perceived risk of corona virus infection across the demographic categories indicate that there are no significant differences across sex but significant differences exist across race and geographical locations (Fig 3). Respondents based in rural locations reported significantly lower risk than those in urban areas. Both locations report increased risk perceptions in the second wave and the difference in the risk perceptions across the geographical divide remains significant. The lower risk perception in rural areas can be attributed to a cognitive assessment based on the lower density of populations in relation to urban areas and lesser interaction with the outside world, which lowers the probability of acquiring this "imported" virus.

An analysis of risk perceptions levels across the income quintiles is further revealing (Fig 4). There are significant differences in risk perceptions across income groups with higher income quantiles having significantly higher risk perceptions compared to the lower income quantiles. Explaining this rationally is difficult considering that higher income groups are in a better position to adopt protective measures. However, over exposure to information, especially from social media and informal sources, can contribute to heightened risk perceptions. Further, an

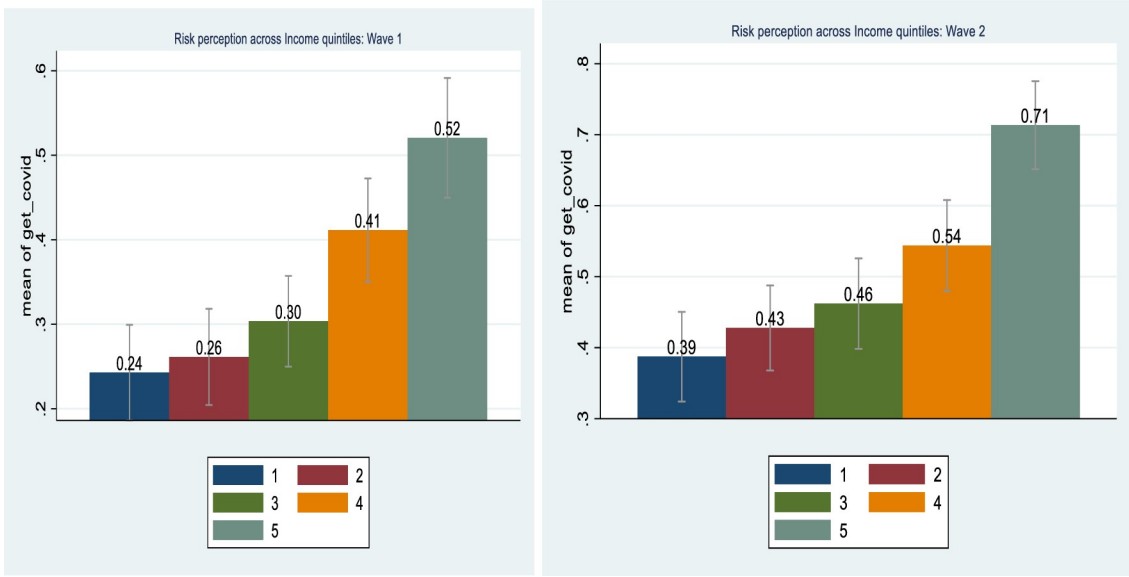

**Fig 4. Risk perception across income quintiles.** Source: NIDS-CRAM, Wave 1 (2020) and NIDS-CRAM, Wave 2 (2020). Data are weighted. The bars show the risk perception across income quintiles. 95% confidence intervals are shown.

emotional response not entirely grounded in reality can contribute to higher risk perceptions [30]. Contrary to this, there is a distinct optimism bias among the lower income quantiles, although this reduced from Wave 1 to Wave 2.

Further, we use the concentration index to quantify the level of concentration of risk perception along key continuous variables like income, education and age. The concentration index is a measure of socio-economic inequality based on the ranking of individuals by some measure of socio-economic status. This paper uses household per capita income, education and age as the ranking variables so that the risk perception levels of individuals can be compare across the levels of these variables [33]. Given that the risk perception variable is binary, we estimate the Erreygers'-corrected concentration index [34]. The results in Table 5 indicate significant pro-rich, pro-education and pro-age concentrations of risk perception. This implies that risk perception is concentrated more among the richer, more educated and older respondents.

Although age is considered to be a factor in the severity of symptoms, hospitalisation and fatality, it is not clear that infection itself can be differentiated along age lines among adults. According to the Centre for Disease Control and Prevention (CDC), the age group 18–27

**Table 5. Erreygers'-corrected concentration index of risk perception.**

| Concentration index | Wave 1 | Lower Bound CI | Upper Bound CI | Wave 2 | Lower Bound CI | Upper Bound CI |
|---|---|---|---|---|---|---|
| Income-related | 0.231*** | 0.161 | 0.301 | 0.240*** | 0.179 | 0.299 |
| Education-related | 0.134*** | 0.085 | 0.183 | 0.145*** | 0.093 | 0.196 |
| Age-related | 0.071*** | 0.021 | 0.122 | 0.103*** | 0.048 | 0.158 |

Source: NIDS-CRAM, Wave 1 (2020) and NIDS-CRAM, Wave 2 (2020)

Notes: Data are weighted. CI is the confidence interval.

*** $p < 0.01$

** $p < 0.05$

* $p < 0.1$

years had three times the risk of infection compared to the 5–17 year group. The older age groups (27+ years) in comparison had only two times the risk of infection of the 5–17 year group, indicating the young adult group (18–27) to be at higher risk of infection [35]. Studies further indicate that Covid-19 transmission through young adults is higher compared to other age groups [36]. Therefore, the inference is that the youth suffer from an optimism bias compared to the elderly.

It is of concern that the concentration levels are high and are increasing along income, education and age factors across waves. The highest concentration of risk perception is along income lines, highlighting the social and health implications of the income divide of the country. Given that the black and rural populations in South Africa have lower average incomes [5, 6], the lower risk perception identified earlier among the black and rural populations could be driven by income as the confounding factor. Isolating the impact of various variables therefore calls for a multivariate analysis, which is undertaken in Section 4.

### 3.4 Self-efficacy

The belief that positive health outcomes can be achieved through personal action (self- efficacy) is an important motivation for individual good health behaviour [37]. Self-efficacy is measured in NIDS-CRAM data through the 'yes' or 'no' response to the question, 'Can you avoid getting the corona virus?'. Self-efficacy, unlike risk perception, has remained unchanged over the two waves, at 87% (Fig 5). Self-efficacy is significantly higher among the majority black African population (accounting for 82% of the country's population), compared to the minority non-black population. The rural population that accounts for one-third of the country's population accounts for higher self-efficacy compared to their urban counterparts.

## 4. Multivariate analysis of health-response behaviour

Although the above bivariate analysis gives interesting insights, it does not control for confounding factors driving relationships, causing possible misinterpretations. Multivariate

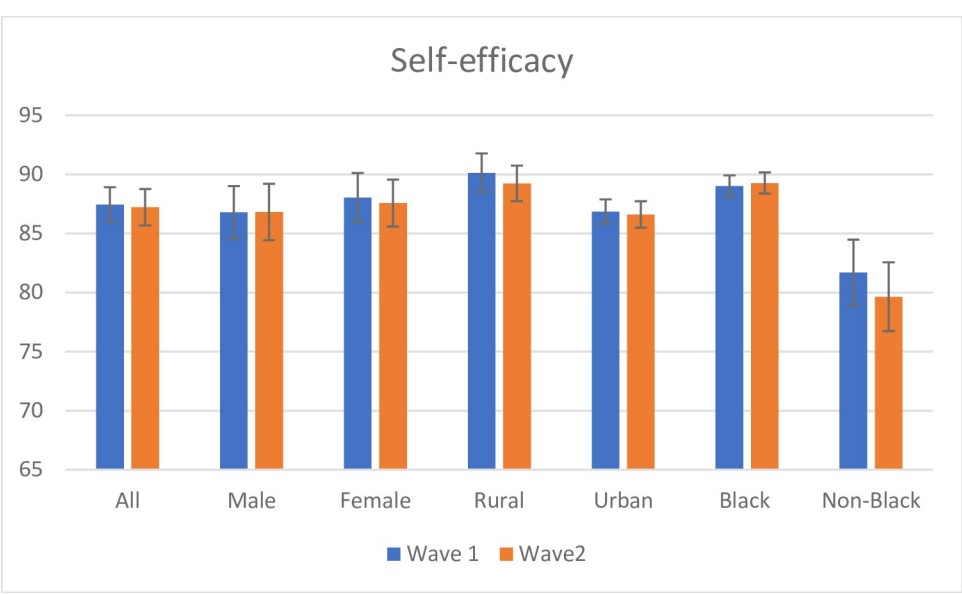

**Fig 5. Self-efficacy across Wave 1 and Wave 2.** Source: NIDS-CRAM, Wave 1 (2020) and NIDS-CRAM, Wave 2 (2020). Data are weighted. The bars show the reported self-efficacy. 95% confidence intervals are shown.

analysis allows one to address this issue and is undertaken in two steps. First, the drivers of behavioural response to the pandemic are estimated within the context of the health belief model. Multiple estimation techniques are utilised to account for possible limitations of available techniques in the context of possible simultaneity between risk perception and preventive behaviour, discussed in Section 2. Second, in order to identify the specific nature of behavioural change, a seemingly unrelated regression model is estimated between the most prominent behavioural change strategies, viz., the use of face masks, handwashing, sanitisers, social distancing and staying at home.

## 4.1 Behavioural change

Given the binary outcome variable on behavioural change (taking the value 1 for those who changed behaviour in some manner or the other and 0 for those who have not changed behaviour), a probit regression estimation is an appropriate starting point. However, considering the possible endogeneity arising through simultaneity between risk perception and behavioural change, we introduce the control function approach proposed by [38] using the *ivprobit* module in STATA. The control function approach is estimated with neighbourhood conditions as instruments. The Wald test of endogeneity is not rejected (p value: 0.6753). However, this result is not sufficient to revert to the baseline probit model because, although the control function approach accounts for the endogenous regressor, it is not appropriate for non-linear models and when the endogenous regressor is discrete [39]. Therefore, we proceed to use the special regressor method estimation [40, 41] to counter any possible bias due to the binary nature of both our dependent variable as well as the endogenous regressor. In order to estimate the special regressor model, a special regressor (V) satisfying the assumption conditions that a) it is continuously distributed and has a large support; b) it is exogenous; and c) it is conditionally independent of the model error term. We chose the age of the individual as V since it is continuously distributed with large support (varying from 17 years to 101 years). However, a limitation of the special regressor method is that the large support condition is not testable [42].

The special regressor-based regression is estimated using the *sspecialreg* module in STATA [43]. The classical tests of instruments' validity can be applied at the final two-stage least squares regression [44]. In our estimation, risk perception is instrumented with two neighbourhood variables that explore the adherence to government regulations (The variables are based on the survey questions: "*How many people in your neighbourhood, if any, went out and drank alcohol with their friends during lockdown?*" and "*How many people in your neighbourhood stayed home and did not go out for social activities or to see their family?*", with the response options provided as: "*None, A few people, About half, Most people*"). The Wu-Hausman test of endogeneity is rejected (p-value: 0.0012), indicating possible bias in the baseline estimations. The Sargan-Hansen test of over-identification (p-value: 0.6574) were performed at the last stage of the special regressor model estimation and confirmed the validity of the two instruments.

The results indicate significant upward bias among the baseline estimations in relation to the special regressor regression result (Table 6). The health belief model is validated with both risk perception as well as self-efficacy variables being positive and significant predictors of behavioural change across the three estimations. The former has a stronger effect on behavioural change. Perceived awareness is also found to be an important correlate, with reporting no source of reliable information being negatively associated with behavioural change. Household income is a positive correlate of behavioural change, as expected, indicating the feasibility and barriers to adoption of behavioural change are an important driver as set out in the health belief model.

**Table 6. Multivariate analysis (dependent variable: Behavioural change).**

| VARIABLES | (Probit)<br>Marginal Effects | (IV Probit)<br>Marginal Effects | (Special regressor)<br>Marginal Effects |
|---|---|---|---|
| Risk perception | 0.163** | 1.984*** | 0.0245*** |
| | (0.0751) | (0.118) | (11.09) |
| Efficacy | 0.353*** | 0.225*** | .0041*** |
| | (0.0883) | (0.0687) | (1.702) |
| Education | 0.0375*** | .04002*** | .00149*** |
| | (0.00942) | (0.000) | (0.000) |
| Household size | -0.00338 | -0.0102 | -.00003 |
| | (0.0113) | (0.00673) | (0.184) |
| Employed | 0.0800 | 0.174*** | 0.0031*** |
| | (0.0752) | (0.0517) | (0.00018) |
| Household income | 0.0516*** | 0.0143 | .00051*** |
| | (0.0152) | (0.0137) | (0.0000) |
| Male | -0.0543 | -0.0129 | .00062164* |
| | (0.0697) | (0.0416) | (.806) |
| Black African | 0.689*** | 0.447*** | .00372411*** |
| | (0.0891) | (0.0865) | (1.753) |
| Urban | 0.0967 | -0.0500 | -.00020591 |
| | (0.0805) | (0.0505) | (1.229) |
| Self reported source of reliable information | | | |
| Government | 0.0566 | 0.0382 | 5.390e-06 |
| | (0.106) | (0.0622) | (1.581) |
| News | 0.232*** | 0.166*** | 0. 00028789 |
| | (0.0859) | (0.0570) | (1.421) |
| Health worker | 0.0183 | 0.1000 | -.00036339 |
| | (0.103) | (0.0611) | (1.741) |
| Community leader | -0.267 | 0.00763 | -.00194134 |
| | (0.173) | (0.119) | (3.355) |
| Social media | 0.173 | 0.0348 | .00252453*** |
| | (0.117) | (0.0731) | (1.743) |
| Acquaintance | -0.114 | -0.295*** | .00007798 |
| | (0.173) | (0.106) | (3.253) |
| No source | -0.562** | -0.0272 | -.00490862** |
| | (0.283) | (0.223) | (0.000) |
| Time | 1.305*** | 0.239 | .00786*** |
| | (0.123) | (0.188) | (0.000) |
| Constant | -1.666*** | -0.316 | -.00576327 |
| | (0.300) | (0.214) | (4.614) |
| Observations | 5501 | 5501 | 5364 |

Standard errors in parentheses

*** $p<0.01$

** $p<0.05$

* $p<0.1$

In addition, there are significant socio-economic drivers behind the health-response behaviour. It is clear through all the models that education plays a significant role in driving behaviour. As expected, those employed are positively correlated with some form of behavioural

change. The majority black African population are more likely to have made some behavioural change compared to the minority population groups. Similarly, males are significantly more likely to have changed their behaviour compared to females. It is clear from the findings that, in addition to the Health Belief Model, socio-economic drivers are relevant in moulding the response behaviour in South Africa.

Considering that both the survey waves included in this study were undertaken before the pandemic peaked in the country, the increasing trend observed in behavioural-change adoption is in line with the growing awareness and anxiety among the population.

## 4.2 Preventive behaviour

While the findings in the earlier section validate the Health Belief Model and broadly match that of the earlier studies undertaken in the context of the SARS pandemic [9], a deeper understanding of the socio-economic correlates of the type of behavioural change is warranted. Behavioural change in the form of preventive strategies, as discussed in earlier sections, have primarily taken the form of the use of face masks, handwashing, use of sanitisers, social distancing and staying home. We next look at the socio-economic drivers of preventive behaviour for each of these preventive strategies. A seemingly unrelated regression [45] is the appropriate estimation method, considering the correlation across the error terms of the various preventive strategy regressions.

It is most interesting to see a clear shift in preventive behaviour strategy over time (Table 7). There is increased use of face masks and hand-sanitisers, while other preventive strategies mark a significant decline. This is in keeping with the gradual restarting of economic activity, making staying home and social distancing difficult. Also, with a better understanding of the airborne nature of the infection, the public messaging focus from government and media shifted from staying home, social distancing and handwashing to the use of sanitisers and face masks.

Despite face-mask use having increased significantly across the population spectrum, there is a positive and significant association in its use among the educated, employed and those who perceive access to some source of reliable information. Age has a non-linear relationship with the use of face masks, handwashing and social distancing strategies. Apparently, the middle aged are the most compliant in these practices, perhaps because the working age group are most exposed to the risk of infection.

Handwashing has declined over time and is practised as a strategy more by females, black Africans, those living in larger households, and with perceived access to reliable information. The use of hand sanitisers has increased over time, with females, urban residents and persons with higher socio-economic status more likely to use hand sanitisers. With increased mobility outside of homes, the use of hand sanitisers seems to have replaced handwashing as a more practical mode of hand hygiene.

Gender is emerging as a key driver of preventive strategies. Males are less likely to adopt stay home, sanitisers and handwash strategies compared to women. However, they are more likely to use the social distancing strategy compared to females. These findings are in line with a study on the practice of social distancing done in the Egyptian context [46]; being male and working and living in urban communities positively contribute to the practice of social distancing, with high statistical significance. In addition, our study indicates that household income and race also are contributing factors to the practice of social distancing.

Education, across the board is positively associated with the use of preventive strategies, except for handwashing. Neighbourhood effects are strongest for social distancing, with non-adherence to government regulations leading to a negative effect on the practice of preventive

**Table 7. Seemingly unrelated regression results (preventive strategy).**

| VARIABLES | Face mask | Handwash | Social distance | Use sanitiser | Stay home |
|---|---|---|---|---|---|
| Age | 0.00647*** | 0.00655*** | 0.00327* | 0.000504 | -0.00317 |
| | (0.00222) | (0.00221) | (0.00189) | (0.00188) | (0.00233) |
| Age squared | -6.21e-05*** | -6.77e-05*** | -3.24e-05 | -1.95e-05 | 3.99e-05 |
| | (2.41e-05) | (2.40e-05) | (2.06e-05) | (2.04e-05) | (2.53e-05) |
| Household income | -0.00356 | 0.00578* | 0.00837*** | 0.00726*** | -0.0121*** |
| | (0.00323) | (0.00321) | (0.00276) | (0.00273) | (0.00339) |
| Employed | 0.0230* | 0.00907 | 0.00801 | 0.0671*** | -0.118*** |
| | (0.0132) | (0.0131) | (0.0113) | (0.0112) | (0.0138) |
| Male | 0.0178 | -0.0807*** | 0.0288*** | -0.0218** | -0.0310** |
| | (0.0126) | (0.0126) | (0.0108) | (0.0107) | (0.0133) |
| Black African | 0.0298 | 0.174*** | -0.0467*** | -0.00744 | -0.0529*** |
| | (0.0192) | (0.0191) | (0.0164) | (0.0162) | (0.0201) |
| Education | 0.00515*** | 0.00113 | 0.0106*** | 0.00453*** | 0.00395** |
| | (0.00178) | (0.00177) | (0.00152) | (0.00151) | (0.00187) |
| Household size | -0.00113 | 0.00755*** | 0.00182 | 0.000942 | -0.00119 |
| | (0.00208) | (0.00206) | (0.00177) | (0.00176) | (0.00218) |
| Neighbourhood non-adherence | -0.00651 | -0.00818 | -0.0211** | -0.00599 | -0.0142 |
| | (0.0125) | (0.0124) | (0.0107) | (0.0106) | (0.0131) |
| Urban | -0.0176 | -0.0126 | 0.0110 | 0.0506*** | 0.00182 |
| | (0.0137) | (0.0136) | (0.0117) | (0.0116) | (0.0144) |
| Self-reported source of reliable information | | | | | |
| Government | 0.0331* | 0.0206 | 0.0404** | -0.0188 | 0.00806 |
| | (0.0184) | (0.0183) | (0.0157) | (0.0156) | (0.0194) |
| News | 0.0348** | 0.0524*** | 0.0581*** | 0.0115 | 0.0126 |
| | (0.0159) | (0.0158) | (0.0136) | (0.0135) | (0.0167) |
| Health worker | 0.0374** | 0.0391** | 0.0411** | 0.00730 | -0.0134 |
| | (0.0190) | (0.0189) | (0.0162) | (0.0161) | (0.0199) |
| Community leader | 0.0764** | 0.0843** | 0.124*** | -0.0523 | 0.0549 |
| | (0.0384) | (0.0382) | (0.0328) | (0.0325) | (0.0403) |
| Social media | 0.0165 | 0.00321 | 0.0558*** | 0.0416** | 0.00487 |
| | (0.0202) | (0.0201) | (0.0172) | (0.0171) | (0.0212) |
| Acquaintance | 0.0997*** | 0.0725** | 0.0342 | 0.0104 | -0.0322 |
| | (0.0342) | (0.0340) | (0.0292) | (0.0290) | (0.0359) |
| No source | -0.0273 | -0.200*** | 0.0546 | -0.00986 | 0.134* |
| | (0.0724) | (0.0720) | (0.0618) | (0.0613) | (0.0761) |
| Wave | 0.204*** | -0.0561*** | -0.0297*** | 0.219*** | -0.0996*** |
| | (0.0119) | (0.0118) | (0.0102) | (0.0101) | (0.0125) |
| Constant | 0.111* | 0.361*** | -0.0419 | -0.254*** | 0.766*** |
| | (0.0671) | (0.0667) | (0.0573) | (0.0568) | (0.0705) |
| Observations | 6238 | 6238 | 6238 | 6238 | 6238 |
| R-squared | 0.057 | 0.035 | 0.035 | 0.099 | 0.037 |

Standard errors in parentheses

*** $p < 0.01$

** $p < 0.05$

* $p < 0.1$

strategies. This is significant for social distancing, given the public good nature of the practice. It requires cooperation from other individuals and cannot be implemented unilaterally by an individual.

As expected, with the easing of lockdown conditions and the reopening of the economy, the employed have not been able to adopt a staying home strategy. Use of face masks and hand-sanitisers are positive correlates of those employed. Per capita household income has a negative association with the staying home strategy. Other strategies, except face masks, are significantly associated with income. This finding highlights the barriers against the adoption of certain strategies. For example, social distancing is not feasible for those without private vehicles and for those who rely on public transport. Similarly, the cost of sanitisers is a barrier for the poor. The use of face masks is across the income spectrum and as such we do not see a significant association between income and face-mask use.

The perceptions on sources of reliable information also throw light on the chosen preventive strategies. Those who perceived news, community leaders and health workers as sources of reliable information were more likely to adopt face masks, handwashing and social distancing strategies. Those who reported no source of reliable information were more likely to be staying home.

## 5. Discussion and conclusion

This study undertakes an in-depth look into the socio-economic inequality of behavioural responses towards the corona virus pandemic. Despite the high income inequality and poverty in the country, the enhanced behavioural response is comparable to that of other developed countries [13]. There has been an increase in enhanced behavioural responsiveness, with 99% (as against 92% earlier) of respondents reporting some form of change in behaviour as a preventive measure against infection. Preventive behaviour is evolving over time; the use of face masks has overtaken handwashing as the most utilised preventive measure. Over 70% of respondents in June indicated the use of face masks, an increase from under 50% in April. Handwashing featured as the second most popular measure in June. While there is an increased use of hand sanitisers and home cleaning as preventive measures against infection in June as compared to April, other measures like social distancing, avoiding close contact, avoiding big groups and staying at home have declined subsequently between the two periods. The increasing adoption of face masks and decline in social distancing are in line with the trends observed in developed countries [16]. This underlines the need for public messaging to emphasise the complementary nature of these measures, given that any one measure in itself is not sufficient on its own.

Our findings are aligned with [15] who found older and more educated adults had higher risk perceptions compared to young and less-educated adults. In addition, we find that risk perception is significantly concentrated among the higher income groups. Despite higher vulnerability, there is an optimism bias among black South Africans, lower income, less educated and younger age groups. As Covid-19 vulnerability is observed to be associated with multidimensional poverty [32], the perceived differences in risk appears to be the result of two possible biases: optimism bias among the less affluent (the more vulnerable category) and an over estimation of risk among the affluent (less vulnerable) sections. This points to the continued tag of Covid-19 as a "rich man's disease" [47–49]. The self-efficacy rate has remained unchanged with 87% of respondents reporting that Covid-19 can be avoided in both survey periods.

The multivariate model that accounts for endogeneity validates the Health Belief Model. Perceived risk, self-efficacy, perceived awareness and barriers to preventive strategy adoption

are significant drivers of health-response behaviour. Social economic factors also play an important role in this regard. It is clear that, with the opening up of the economy and the return of individuals to employment; it has become harder for individuals, especially in the lower income categories, to observe physical distancing. There is significant income- and education-related inequality between the types of preventive measures adopted. Measures such as social distancing, avoiding close contact, and the use of sanitisers are practised more by the rich and educated. The low-income respondents are not able to maintain physical distancing measures as the economy opens up. This highlights the question of the feasibility of these preventive measures for the poor who live in crowded households and neighbourhoods and have no alternative to public transport. The lifting of capacity restrictions in public mini-bus taxis have particular bearing for individuals from the lower economic strata of society, exposing them to higher risks compared to those with private transport. It is recommended that the government reintroduce capacity restrictions in public transport to protect the vulnerable who do not have access to private transportation.

This study highlights the need to consider the individual motivation and impediment factors as additional drivers of behavioural response. The feasibility of adopting a certain preventive measure by an individual is contingent on their living and livelihood circumstances. The awareness campaigns and policy recommendations therefore have to talk to the lived realities of individuals in different circumstances. Practical interventions, like making sanitisers freely available in public spaces where people tend to congregate, making free face masks available to the poorest of the poor, among others, are recommended. Moreover, the optimism bias recorded in the literature [50] can lead people to risky behaviour because they falsely believe that they are less at risk of negative events than are other people. The study has identified the categories of individuals more prone to optimism bias to enable more targeted awareness creation.

## Author Contributions

**Conceptualization:** Umakrishnan Kollamparambil, Adeola Oyenubi.

**Formal analysis:** Umakrishnan Kollamparambil, Adeola Oyenubi.

**Methodology:** Umakrishnan Kollamparambil.

**Writing – original draft:** Umakrishnan Kollamparambil.

**Writing – review & editing:** Adeola Oyenubi.

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
