## [Decision Letter · Decision Letter 0]

25 Feb 2021

PONE-D-20-31994

Behavioral response to Covid19 Pandemic in a high income-inequality country

PLOS ONE

Dear Dr. Kollamparambil,

Thank you for submitting your manuscript to PLOS ONE. After careful consideration, we feel that it has merit but does not fully meet PLOS ONE’s publication criteria as it currently stands. Therefore, we invite you to submit a revised version of the manuscript that addresses the points raised during the review process.

Apart from referees comments, I suggest you to go for substantial language proofreading of the paper for correcting typos, grammar, and syntax. 

We look forward to receiving your revised manuscript.

Kind regards,

Srinivas Goli, Ph.D.

Academic Editor

PLOS ONE

Additional Editor Comments:

Apart from referees comments, I suggest you to go for substantial language proofreading of the paper for correcting typos, grammar, and syntax.

Journal Requirements:

3. We noted in your submission details that a portion of your manuscript may have been presented or published elsewhere.

"The figures and the bivariate analysis included are published in a NIDS-CRAM policy report. However, the Multivariate analysis (probit, control function method, special regressor method and seemingly unrelated regression) is unique to this paper. This paper is framed within the context of behavioral change whereas the policy paper is focussed on risk perception. Therefore, even though the preliminary statistical analysis is the same, this paper adds further value through rigorous econometric analysis accounting for endogeneity and addressing a broader issue. "

Please clarify whether this publication was peer-reviewed and formally published. If this work was previously peer-reviewed and published, in the cover letter please provide the reason that this work does not constitute dual publication and should be included in the current manuscript.

Reviewers' comments:

Reviewer's Responses to Questions

**Comments to the Author**

1. Is the manuscript technically sound, and do the data support the conclusions?

Reviewer #1: Partly

Reviewer #2: Yes

2. Has the statistical analysis been performed appropriately and rigorously? 

Reviewer #1: I Don't Know

Reviewer #2: Yes

3. Have the authors made all data underlying the findings in their manuscript fully available?

Reviewer #1: Yes

Reviewer #2: Yes

4. Is the manuscript presented in an intelligible fashion and written in standard English?

Reviewer #1: Yes

Reviewer #2: No

5. Review Comments to the Author

Reviewer #1: I think that this article first needs a major revision before becoming acceptable for publication. Below, I discuss major and minor comments.

Major comments:

1) Scientific quality is the essence of this journal so please pay careful attention to describing the (validity of) your data. Table 1: please report how the descriptive statistics relate to data in the population. Report whether there are significant differences between the sample and the population and how this affects your results. Also describe measures such as the response rate of the survey. How did you dealt with non-response? It is possible to check differences between slow and fast responders? Slow responders might be a good proxy for non-responders the literature shows. Did your conduct data cleaning and if so how?

2) How did you measure optimism bias exactly. You ask for risk perception, but maybe the risk perception of the respondents is really accurate?

3) The abstract is too lenghty to get the main message. It’s now 446 words. Please try to downsize it to max. 300 words. Cut it to the bone.

4) I think there is more literature on behavioral responses in the COVID-19 crisis. I am quite sure I have recently read an article in this journal about the topic. Please include some more relevant literature and, more importantly, discuss not only that these studies are conducted but also what their insights were and how they relate to your insights.

Minor comments:

1) Check the English. I spotted some mistakes

2) Resonsider the title. Behavioral response to Covid19 Pandemic in a high income-inequality country. In the introduction you speak of an economically and societally divided country. Maybe it’s better to also use that in the title or just mention SA in the title instead of high income-inequality country.

3) Figure 2. Use different type of colours for the two waves.

4) Sometimes you already comment on the results in the results section. Those type of phrases should be removed to the discussion section. E.g. page 11: “It is important for public messaging to emphasize the complementary nature of these measures given that any one measure in itself is not sufficient on its own.”

5) A limitation of your study is that it concerns self-reported data. The answers of the respondents were not consequential. Please discuss hypothetical and/or strategic bias. To which extent do you think that respondents answered in a socially desirable way?

6) Figure 3 should be improved. It is not immediately clear what the y-axis means

7) P,.26. This points to the continued tag of Covid19 as a “rich man’s disease”. Please add a reference here. The disease is not so much known like that in my home country.

Reviewer #2: The authors present an interesting topic regarding to the behavioral response to COVID-19 pandemic in high income-inequality country using the health behaviour model.

The authors have used all the data as the dataset provided. But the most important thing is the discussion section, I think the authors should compare their results to the current literatures and highlight the importance factors associated with the inequality. Also, the authors should interpretate the factors in the real-world settings, this means that the authors should state or recommend some suggestions to protect the COVID-19.

Also, we suggest the authors should have native English speakers to go through the manuscript because the manuscript has many grammar errors.

6. PLOS authors have the option to publish the peer review history of their article (what does this mean?). If published, this will include your full peer review and any attached files.

Reviewer #1: No

Reviewer #2: **Yes: **Dr Xiwen Simon Qin

---

## [Author Response · Author response to Decision Letter 0]

12 Mar 2021

Thank you for the considered comments by the reviewers. The authors have revised the manuscript in accordance with the comments received. Specific response to each comment is provided below.

Reviewer #1: I think that this article first needs a major revision before becoming acceptable for publication. Below, I discuss major and minor comments.

Major comments:

1) Scientific quality is the essence of this journal so please pay careful attention to describing the (validity of) your data. Table 1: please report how the descriptive statistics relate to data in the population. Report whether there are significant differences between the sample and the population and how this affects your results. 

RESPONSE: Additional column on population data (based on 2011 census, which is the last available population data) is now included. The survey sample and population figures are comparable for demographic variables. As expected, the Income variable shows a decline during the pandemic.

Also describe measures such as the response rate of the survey. How did you dealt with non-response? 

RESPONSE: Included a discussion on the response rate of the survey, deriving from:

Kerr, A., Ardington, C., Burger, R. (2020). Sample design and weighting in the NIDS-CRAM survey. Cape Town: SALDRU, UCT. (SALDRU Working Paper No. 267) http://www.opensaldru.uct.ac.za/bitstream/handle/11090/983/2020_267_Saldruwp.pdf?sequence=1

Daniels, R.C., Ingle, K.P., Brophy, T.S.L, (2020) Determinants of attrition in NIDS-CRAM Waves 1 & 2. Cape Town: SALDRU, UCT. (SALDRU Working Paper No. 271)

Branson, N. and Wittenberg, M. (2019). Longitudinal and Cross‐Sectional Weights in the NIDS Data 1‐5. NIDS Technical Paper 9.

It is possible to check differences between slow and fast responders? Slow responders might be a good proxy for non-responders the literature shows. 

RESPONSE: No, unfortunately this information is not available.

Did you conduct data cleaning and if so how?

RESPONSE: The data cleaning was done in Stata. The main step involved identifying the variable necessary for our analysis and dealing with non-responses. Nonresponses (denoted by Negative values in the dataset) were coded as missing and were removed from the analysis. This particularly affect the household income variable and this has been acknowledged in the paper. However, the finding that other measures of socioeconomic status like race and education show similar pattern to income suggest that the result is not biased by the missing observations. 

2) How did you measure optimism bias exactly. You ask for risk perception, but maybe the risk perception of the respondents is really accurate?

RESPONSE: Covid19 vulnerability is related to covariates associated with multidimensional poverty in South Africa (StatasSA 2021). These have been identified as: no access to private transport, lack of access to Covid related information, multigenerational households, overcrowding, lack of water and sanitation facilities, elderly, chronic medication. Except age, the other factors are closely associated with poverty.

http://www.statssa.gov.za/publications/Report%2000-80-05/SACVI%20Technical%20Report.pdf

Therefore, those on the lower income quintiles are more vulnerable to Covid19, yet this category assesses the risk as lower, pointing to optimism bias. Extending this, poverty in South Africa is known to be race-based. Poverty rate of black South Africans are much higher than other race groups (Leibbrandt, F. and Woolard, I. 2010). As such, vulnerability on average can be said to be higher among this race. However, assessment of risk by back South African is lower pointing optimism bias.

A discussion of the above is now included in the paper where optimism bias is indicated.

3) The abstract is too lengthy to get the main message. It’s now 446 words. Please try to downsize it to max. 300 words. Cut it to the bone.

RESPONSE: Now cut down to 300 words.

4) I think there is more literature on behavioral responses in the COVID-19 crisis. I am quite sure I have recently read an article in this journal about the topic. Please include some more relevant literature and, more importantly, discuss not only that these studies are conducted but also what their insights were and how they relate to your insights.

RESPONSE: Included more literature that has emerged since the submission of the manuscript to the journal in October 2020. 

Minor comments:

1) Check the English. I spotted some mistakes

RESPONSE: The manuscript has now gone through professional editing.

2) Reconsider the title. Behavioral response to Covid19 Pandemic in a high income-inequality country. In the introduction you speak of an economically and societally divided country. Maybe it’s better to also use that in the title or just mention SA in the title instead of high income-inequality country.

RESPONSE: changed as recommended

3) Figure 2. Use different type of colours for the two waves.

RESPONSE: Changed colour to differentiate between waves better.

4) Sometimes you already comment on the results in the results section. Those type of phrases should be removed to the discussion section. E.g. page 11: “It is important for public messaging to emphasize the complementary nature of these measures given that any one measure in itself is not sufficient on its own.”

RESPONSE: Removed and relocated from results section to the discussion section.

5) A limitation of your study is that it concerns self-reported data. The answers of the respondents were not consequential. Please discuss hypothetical and/or strategic bias. To which extent do you think that respondents answered in a socially desirable way?

RESPONSE: In order to minimise the pressure to respond in a socially desirable way, all participants in survey were verbally informed that the identity of the respondent would be kept confidential and all the collected information would be anonymised. Further, the respodents were made aware that participation in the study was voluntary, and that they could stop the interview at any time. Despite all of these measures, it is hard to assert that there was no strategic bias. This has been highlighted as a study limitation. It is difficult to ascertain the level of bias from the available data and has been acknowledged. 

6) Figure 3 should be improved. It is not immediately clear what the y-axis means

RESPONSE: Y-axis label included

7) P,.26. This points to the continued tag of Covid19 as a “rich man’s disease”. Please add a reference here. The disease is not so much known like that in my home country.

RESPONSE: In the first months of the pandemic, COVID-19 was sometimes presented as the ‘great equalizer’, in that the people who travelled (and who therefore may have had higher socio-economic status) were initially more likely to be infected. Three references are included.

Reviewer #2: The authors present an interesting topic regarding to the behavioral response to COVID-19 pandemic in high income-inequality country using the health behaviour model.

The authors have used all the data as the dataset provided. But the most important thing is the discussion section, I think the authors should compare their results to the current literatures and highlight the importance factors associated with the inequality. 

RESPONSE: Discussion section now includes comparison of study findings with existing literature on Covid19.

Also, the authors should interpretate the factors in the real-world settings, this means that the authors should state or recommend some suggestions to protect the COVID-19.

RESPONSE: Have included some practical recommendations in the discussion section to protect from Covid19. 

Also, we suggest the authors should have native English speakers to go through the manuscript because the manuscript has many grammar errors.

RESPONSE: The manuscript has now gone through professional editing.

---

## [Decision Letter · Decision Letter 1]

5 Apr 2021

Behavioural response to the Covid-19 pandemic in South Africa

PONE-D-20-31994R1

Dear Dr. Kollamparambil,

We’re pleased to inform you that your manuscript has been judged scientifically suitable for publication and will be formally accepted for publication once it meets all outstanding technical requirements.

Kind regards,

Srinivas Goli, Ph.D.

Academic Editor

PLOS ONE

Additional Editor Comments (optional):

Both reviewers are recommend this article and I go with them.

Reviewers' comments:

Reviewer's Responses to Questions

**Comments to the Author**

1. If the authors have adequately addressed your comments raised in a previous round of review and you feel that this manuscript is now acceptable for publication, you may indicate that here to bypass the “Comments to the Author” section, enter your conflict of interest statement in the “Confidential to Editor” section, and submit your "Accept" recommendation.

Reviewer #1: All comments have been addressed

Reviewer #2: All comments have been addressed

2. Is the manuscript technically sound, and do the data support the conclusions?

Reviewer #1: Yes

Reviewer #2: Partly

3. Has the statistical analysis been performed appropriately and rigorously? 

Reviewer #1: Yes

Reviewer #2: Yes

4. Have the authors made all data underlying the findings in their manuscript fully available?

Reviewer #1: Yes

Reviewer #2: Yes

5. Is the manuscript presented in an intelligible fashion and written in standard English?

Reviewer #1: Yes

Reviewer #2: Yes

6. Review Comments to the Author

Reviewer #1: The authors addressed my comments in a satisfactory way. I think that this paper provides interesting insights regarding behavioural responses to COVID-19 in SA

Reviewer #2: (No Response)

7. PLOS authors have the option to publish the peer review history of their article (what does this mean?). If published, this will include your full peer review and any attached files.

Reviewer #1: No

Reviewer #2: No

---

## [Editor Report · Acceptance letter]

8 Apr 2021

PONE-D-20-31994R1 

Behavioural response to the Covid-19 pandemic in South Africa 

Dear Dr. Kollamparambil:

I'm pleased to inform you that your manuscript has been deemed suitable for publication in PLOS ONE. Congratulations! Your manuscript is now with our production department. 

Kind regards, 

on behalf of

Dr. Srinivas Goli 

Academic Editor

PLOS ONE